# Spatial and Temporal Variation and Prediction of Ecosystem Carbon Stocks in Yunnan Province Based on Land Use Change

**DOI:** 10.3390/ijerph192316059

**Published:** 2022-11-30

**Authors:** Long Li, Wei Fu, Mingcan Luo

**Affiliations:** School of Economics and Management, Southwest Forestry University, Kunming 650233, China

**Keywords:** Yunnan Province land use change, InVEST model, CA-Markov model, ecosystem carbon stock

## Abstract

The function of ecosystems as carbon sinks has emerged as a key strategy for advancing the concept of “carbon neutrality” and “carbon peaking”. Ecosystem carbon stocks are impacted by land use changes that alter ecosystem structure and function. We evaluated the ecosystem carbon stocks of Yunnan Province in different periods with the aid of the carbon stock module of the InVEST model, analyzed the relationship between land use type shift and ecosystem carbon stock changes, and combine them with the CA-Markov model to predict land use types in 2030. The results showed that between 1990 and 2020, changes in land use primarily affected cropland, grassland, and forested areas. The ecosystem’s average carbon stock from 1990 to 2020 was 8278.97 × 10^6^ t. The carbon stocks of cropland, grassland, and unused land decreased by 31.36 × 10^6^ t, 32.18 × 10^6^ t, and 4.18 × 10^6^ t during 1990–2020, respectively, while the carbon stocks of forest land, water area, and construction land increased by 24.31 × 10^6^ t, 7.34 × 10^6^ t, and 22.08 × 10^6^ t. The main cause of the increase in carbon stocks in the ecosystem in Yunnan Province throughout the process of land use type shift was the development of forest land area, whereas the main cause of the decline was the shrinkage of cropland and grassland areas.

## 1. Introduction

The destruction of the natural ecological environment brought about by global rapid economic growth and the effects of human behavior have significantly altered the global climate, making greenhouse effect-driven climate change one of the main issues of concern for the entire international community [1]. Carbon emission reduction has become a global consensus, and nations throughout the world have made a deliberate effort to reduce their carbon emissions as a result of the immediate harm that increasing CO_2_ concentrations in the atmosphere have caused to human habitation. The reduction of greenhouse gas emissions has become a global priority [2]. China is now the world’s greatest emitter of CO_2_, according to the International Energy Agency’s (IEA) statistics on worldwide energy and CO_2_ emissions [3]. China’s future development will increasingly focus on carbon emission reduction and transformation development [4]. General Secretary Xi made the “Dual carbon” target explicit for the first time during the United Nations General Assembly’s 75th session. The majority of past studies on CO_2_ emissions have concentrated on carbon sources rather than the function of carbon sinks [5]. To continuously advance the reduction of greenhouse gas emissions, we must balance the two factors in the emission reduction process and not just concentrate on “carbon sources” but also on “carbon sinks.” Global terrestrial ecosystems absorb 31% of anthropogenic CO_2_ emissions between 2010 and 2019 [6,7]. Terrestrial ecosystems, as a significant component of the global ecosystem, play a critical role in lowering CO_2_ emissions. Ecosystem carbon sinks are essential for achieving carbon neutrality in the context of the “double carbon” target, so they must be given special consideration. The more carbon sinks there are, the more likely it is that carbon neutrality will be reached [8].

Dramatic changes in land use and cover (LUCC) have an impact on terrestrial ecosystems’ ability to function as well as their ability to store carbon [9,10,11]. The worldwide ecosystem carbon cycle, the drop in CO_2_ concentrations, and global climate change are all significantly impacted by changes in terrestrial ecosystem carbon stores [12]. The increase of carbon storage in terrestrial ecosystems is currently one of the most effective and environmentally friendly energy conservation strategies to combat climate change and the greenhouse effect, as doing so will lower atmospheric CO_2_ levels and play a significant role in climate change mitigation. The role of vegetation in carbon sequestration and oxygen release is more prominent in terrestrial ecosystems [13] where vegetation performs photosynthesis and respiration to maintain the dynamic balance of CO_2_ and O_2_ in the atmosphere, and its distribution characteristics can directly affect the changes in carbon storage [14]. The carbon sink service function is affected by complicated climate conditions, plant cover status, and land use patterns, which results in diverse response mechanisms [15]. For instance, as cities grow, grasslands and forests are inevitably converted into construction sites, releasing the aboveground biomass of plants into the atmosphere as a source of carbon dioxide (CO_2_) [16]. Taking measures like converting farmland back to grassland or forest (and vice versa) will increase the carbon sink potential of various land use types [17]. Land use types can function as both carbon sources and sinks due to the various impacts that land use patterns and development have on ecosystem carbon sinks [18]. In order to ameliorate natural circumstances and activate ecosystems’ carbon sink service functions through human intervention, ecological conservation and restoration projects have therefore emerged as one of the most crucial challenges in China’s ongoing effort to adapt to climate change. Therefore, one of the primary methods to sequester carbon and increase sinks in the future is to prioritize protecting grassland and forests in areas with high carbon sink capacity, rationally planning the pattern of use for cropland and construction land, enhancing the function of forests and vegetation in sequestering carbon, and scientifically planning land use patterns [19].

Understanding the effects of changing land use types on carbon stocks in terrestrial ecosystems is essential for better promoting the carbon cycle in these systems. Numerous academics have looked at it from various angles up to this point, with methodologies based on statistical and remote sensing models being used [20,21,22,23]. The bookkeeping model, which excels at tracking various land use types and areas of change in carbon density and is frequently utilized in regional or global ecosystems [24,25,26], has been widely used to compute carbon changes produced by LUCC. For instance, Houghton et al. [12] utilized a bookkeeping model to calculate the carbon emissions brought on by changes in land use in terrestrial ecosystems across the globe between 1850 and 2000. Recent decades have seen a significant amount of research on changes in carbon stocks linked to LUCC in terrestrial ecosystems [27,28,29]. Models, however, typically rely on time-fixed inventory data and the assumption that carbon density doesn’t change over time. In other words, they prioritize comparing the carbon density of various land uses and its static spatial fluctuation, but they neglect its temporal variability. The findings of the bookkeeping model cannot be directly compared with the observed results since the applied function is static [30,31]. The Integrated Valuation of Ecosystem Services and Trade-offs (InVEST) model has been extensively utilized and has steadily evolved into a more conventional methodology to address the drawbacks of bookkeeping models [32,33,34]. Numerous studies have been carried out in the Americas, Europe, Africa, and other significant countries and areas using the InVEST model, which can quantify regional carbon stocks and the impact of land use change on carbon stocks [35,36,37,38]. The InVSET model has also demonstrated remarkable relevance in research on regions in China. For instance, Zhao et al. [34] examined the geographical and temporal variations of the carbon stock in the study and utilized the InVEST model to account for the carbon stock in the Heihe River Basin, the relative error of measurement is only 0.22%. In the Inner Mongolian Xilingol and Qi River basin of Taihang Mountains, Zhang et al. [39] and Zhu et al. [40] assessed carbon stocks, and the results simulated in accordance with the InVEST model were extremely similar to the earlier results of carbon stocks using field sampling. It showed the InVEST model’s applicability and precision.

The aforementioned research has shown the benefits of combining land use type transfer and InVEST models in the empirical analysis of ecosystem carbon stocks. However, there are still several shortcomings in the previous studies on ecosystem carbon stocks, such as the absence of uniform standards for evaluating ecosystem carbon stocks, there aren’t enough studies conducted in southwest China, where there is a significant amount of forest cover, and there are not sufficient predictions of how things will develop in the future. As a result, the InVEST model was used in this study to estimate the ecosystem carbon store in Yunnan Province and to assess its spatial and temporal changes in relation to LUCC changes. The CA-Markov model was used to forecast the land use in Yunnan Province in 2030 based on the 1990–2020 LUCC. By examining the various factors that contribute to changes in land type, the CA-Markov model can simulate various land use changes with accuracy. The Yunnan ecosystem’s pattern of changing carbon stock in 2030 is then predicted. In this study, we measured the ecosystem carbon stocks in Yunnan Province over various timescales, investigated intrinsic mechanisms and mechanisms of land use changes affecting ecosystem carbon stocks based on trends of both ecosystem carbon stocks and land use patterns, and discovered the contribution of various land use types to the changes of ecosystem carbon stocks. We also analyzed the spatial and temporal patterns of changes in ecosystem carbon stocks in different periods, and predict the future changes in land use patterns and ecosystem carbon stocks in Yunnan Province. It effectively monitors changes to the ecosystem in Yunnan Province, used areas with high carbon sink capacities as the main protection objects of carbon sink space, protected grassland and forest land, carefully planned the use of cultivated land and construction land, increased the efficiency of vegetation’s ability to fix carbon, provides data support for the overall spatial planning of Yunnan land, and provides a scientific basis for the spatial planning of land under the goal of “double carbon”.

## 2. Materials and Methods

### 2.1. Overview of the Study Area

Geographically speaking, Yunnan Province is in southwest China and covers 394,100 km^2^. With high topography in the northwest and low terrain in the southeast, the province is primarily mountainous and forested, with a subtropical and tropical monsoon climate. With 259,944 km^2^ of woodland, Yunnan Province possesses China’s second-largest forest acreage and forest stock (Figure 1). Yunnan Province is known for its abundance of greenery, and its ancient forests are home to a wide variety of unique plants and animals. It also has a big number of national forest parks, and the development of an ecological civilization is one of the nation’s top priorities. Yunnan Province has a significant capability for forest carbon sequestration due to the size of its forested area and the support of its forestry policies. Land use changes can have a significant impact on the ecosystem’s carbon store in Yunnan Province because of the province’s distinctive land use pattern, which includes a high share of forest and woodland.

### 2.2. Data Sources

This study examined data on land use and cover for the years 1990, 2000, 2015, 2018, and 2020. The selection of time points was based on the data that was available and the trajectory of land change during the previous ten years. The maps were all created manually by visual interpretation at the Resource and Environment Science and Data Center of the Chinese Academy of Sciences, with an overall accuracy of more than 93%. According to the China Land Resources Classification System’s standards and a geographical resolution of 1 km, the land use types are classed in this study into cropland, forested land, grassland, water area, construction land, and unused land. The vector data of the Yunnan administrative boundary used in the study were obtained from the National 1:1,000,000 Basic Geographic Database published by the National Basic Information Center.

Carbon density statistics, or the amount of carbon stored per unit area in each land use carbon pool, are needed to calculate ecosystem carbon storage in Yunnan Province. It primarily consists of carbon pools made up of aboveground biomass, belowground biomass, soil, and dead organic matter. Carbon density data were obtained mainly by reviewing the relevant literature. First, nationwide carbon density data were obtained based on a study of nationwide carbon density by Li et al. [41]. The worldwide vegetation aboveground biomass dataset [42] and the literature [43] were also included in the aboveground biomass carbon density data. The carbon density of belowground biomass and soil carbon density data referred to global soil carbon density data and the literature [44]. Then, locations that shared the same classification system or similar natural circumstances as Yunnan Province were chosen as comparators [45,46,47,48]. The carbon density values must also be adjusted because they are intimately tied to soil characteristics and types of land use. The formula in the study of Alam et al. [49] was chosen for the correction of the precipitation factor and the formula in the study of Giardina et al. [50] was chosen for the correction of the annual mean temperature and biomass carbon density based on the national and Yunnan multi-year mean temperatures (7.56 °C, 14.5 °C), and precipitation (673.9 mm, 1180.3 mm). Finally, the Yunnan Province’s carbon density information was compiled (Table 1).

### 2.3. Methodology

#### 2.3.1. Carbon Module of InVEST Model

The InVEST model (Integrated Valuation of Ecosystem Services and Trade-offs) is a free and open source model developed with the support of the Natural Capital Project to quantify the functions of multiple ecosystem services. The InVEST model’s carbon module, which primarily consists of four fundamental carbon pools—aboveground biomass carbon pool, belowground biomass carbon pool, soil carbon pool, and dead organic matter carbon pool—uses the LUCC type map and the carbon storage in each carbon pool to estimate the net storage of regional carbon over time. The method of calculation is to calculate the carbon density of each land class based on the average carbon density of the four primary carbon pools of the various land classes, multiply each land class’s carbon density by the area and add the results, and then calculate the total carbon stock of the study area, abbreviated *C_total_*. The calculation formula is as follows:(1)$$C_{i}=C_{above}+C_{below}+C_{soil}+C_{dead}$$
(2)$$C_{total}=\sum_{i=1}^{n}C_{i}*S_{i}$$
where: *i* represents land use type; *C_above_*, *C_below_*, *C_soil_*, and *C_dead_* represent aboveground, belowground, soil, and dead organic matter carbon density, respectively; *C_i_* is the carbon stock of a land type; *C_total_* is the total ecosystem carbon stock; *S_i_* is the area of land use type *i*, and n is the number of land use types.

#### 2.3.2. CA-Markov Model

The stochastic process research of mathematician Markov is where the idea of the Markov prediction originated. The evolution of land patterns is now frequently studied using the Markov prediction principle. In the study of land cover evolution, it is feasible to compare the land use category of one period with the Markov process’s potential outcomes, which are only related to the land use category of the preceding period. Cell Automata (CA) and the Markov model are combined to create the CA-Markov model. When doing intricate spatial simulations where nearby spatial cells and their own cellular properties interact, the CA model performs exceptionally well [51]. The CA model is powerful for spatial operations and discrete in both time and space. This paper used IDRISI software to run the CA-Markov model to predict land use in Yunnan Province in 2030. The following is the representation equation:(3)$$S_{t+1}=f\left(S_{t},N\right)$$
where *S* stands for a finite and discrete set of cells, *t* and *t* + 1 for different moments, *N* for the cellular neighborhood, and f for the local space cellular transformation rule. The LUCC prediction incorporates transitions to the following period based on the transformation probability of the preceding period as well as land use changes between the time periods *t*_1_ and *t*_2_. The Markov chain itself is a stochastic hierarchy, and the Markov transfer matrix is employed to compute the transition probability in order to investigate the likelihood that the LUCC would change over time [52]. Here is the expression formula:$$P_{ij}=\left[\begin{array}{ccc} P_{11} & \cdots & P_{1n}\\ \vdots & \ddots & \vdots \\ P_{n1} & \cdots & P_{nn} \end{array}\right]$$
(4)$$0\leq P_{ij}\leq 1,\sum_{j=1}^{n}P_{ij}=1\left(i,j=1,2,\ldots ,n\right)$$

The type of land use in year *n* is represented by *n*. The aforementioned matrix is computed as a function of *k* if the transfer probability varies. The study used the actual 2020 LUCC of Yunnan Province as its base map, and then applied the CA-Markov model to simulate the 2020 LUCC simulation map based on the 1990 and 2000 LUCC. The accuracy of the simulation was then confirmed using the Crosstab module of IDRISI software. The dataset’s parameters were adjusted to model Yunnan Province’s land use patterns for 2020 and 2030, respectively, and to compare those simulations to Yunnan Province’s actual land use patterns for 2020. Finally, using the ideal parameters, a simulation of the LUCC land use pattern in Yunnan Province in 2030 was run.

## 3. Results

### 3.1. Land Use Structure Change

Cropland, forest, and grassland make up the majority of the land uses in Yunnan Province, making up 97% of its total area. Over 57% of the land used in Yunnan is comprised of them, and primarily of forest land. Water, construction land, and unused land make up a smaller portion of the total area—only approximately 3% (Table 2). In Yunnan province, the extent of various land uses altered. From 1990 to 2020, and the type of land use changed. From 1990 to 2000, the area of each category was ranked as follows. Forest land > grassland > cropland > water area > unused land > construction land, from 2015 to 2020, the area of different types of land use in Yunnan Province was forest land > grassland > cropland > construction land > water area > unused land. Between 1990 and 2020, the area of cropland, grassland and unused land decreased by 2266 km^2^, 1737 km^2^, and 565 km^2^, respectively, while the area of forest land, water area and construction land increased by 942 km^2^, 1033 km^2^, and 2987 km^2^, respectively, with the most obvious increase in the area of construction land, which increased by more than 150% in 30 years. Compared with 1990, construction land has expanded by more than 1.5 times, and the area of construction land exceeds that of water area between 2015 and 2020, accounting for more than 1% of the total area. The rapid growth of construction land shows the changing land use pattern in Yunnan Province.

### 3.2. Land Use Type Shifts

Forest land, grassland, and cropland dominated the change in land use in Yunnan Province from 1990 to 2020 (Table 3). Forest land, grassland, and cropland collectively made up 57.38%, 22.44%, and 17.54% of the transferred out area, while the remaining land use groups made up only 2.64% of the transferred out area. In terms of transferred in area, forest land, grassland, and agriculture made up the majority of transferees, accounting for 57.20%, 22.91%, and 18.14% of it, respectively. Among the six land types, grassland, cropland, and unused land have achieved net outflow, with a net outflow area of 1812.90 km^2^, 2293.73 km^2^, and 561.96 km^2^ respectively, while construction land, forest land, and water area have achieved net inflow, with a net inflow area of 2983.48 km^2^, 689.30 km^2^, and 995.81 km^2^ respectively. Forest land, grassland, and cropland are the principal land types. 32.07% of the cropland that was transferred out was converted to forest land, 17.43% to grassland, and 4.75% to other property. 32.75% of the grassland was converted to forest land, 14.45% to cropland, and 2.16% to other land during this process. 12.8% of forest land was converted to grassland, 9.69% to cropland, and 0.91% to other land throughout this period. Figure 2 and Table 3 both display the specific transfer.

### 3.3. Changes of Ecosystem Carbon Stocks

Using the carbon stock module of the InVEST model, the carbon stocks of Yunnan Province in 1990, 2000, 2015, 2018, and 2020 were computed (Figure 3). At five stages between 1990 and 2020, the ecosystem’s total carbon stock in Yunnan Province was 8284.23 × 10^6^ t, 8279.59 × 10^6^ t, 8288.16 × 10^6^ t, 8272.62 × 10^6^ t, 8270.24 × 10^6^ t. From 1990 to 2000, the carbon stock decreased by 4.64 × 10^6^ t. From 2000 to 2015, carbon stock increased by 8.56 × 10^6^ t, and from 2015 to 2020, carbon stock decreased by 17.92 × 10^6^ t. Between 1990 and 2020, the overall carbon stock maintained a basically flat state, and the overall carbon stock decreased by 13.99 × 10^6^ t.

### 3.4. Carbon Stock Changes by Land Use Type

The amount that various land types add to the Yunnan ecosystem’s carbon pool varies. Cropland, forest land, and grassland make up more of the ecosystem’s total carbon stock, contributing 11%, 68%, and 19% of it, respectively. Forest land use type makes up nearly 70% of it, showing that the function of forest vegetation in carbon sequestration is more apparent. Water area, construction land, and unused land make up less than 3% of the ecosystem’s total carbon stock (Table 4).

In Yunnan Province, land use types changed between 1990 and 2020 as a result of ecological protection policies, economic development, and other anthropogenic activities. The ecosystem’s overall capacity to sequester carbon remained largely unchanged, but the changes in carbon stocks varied depending on the land type. The carbon stocks of cropland, grassland, and unused land decreased by 31.36 × 10^6^ t, 32.18 × 10^6^ t, and 4.18 × 10^6^ t, accounting for 46.31%, 47.52%, and 6.17% of the total carbon reduction, respectively. Among them, the carbon stock of grassland declined most obviously and in a continuous downward trend, and the contribution of grassland to the carbon stock of the ecosystem decreased from 19.62% in 1990 to 19.27% in 2020. Between 1990 and 2020, cropland’s contribution to the ecosystem’s carbon stock declines, from 11.59% in 1990 to 11.23% in 2020. Unused land had little effect on ecosystem carbon stock, making up only 0.14% of the province’s total carbon stock in 2020. It did not change between 1990 and 2000, and decreased between 2010 and 2020.

The carbon stock of forest land, water area, and construction land increased by 24.31 × 10^6^ t, 7.34 × 10^6^ t, and 22.08 × 10^6^ t in order, among which forest land had the largest increase, accounting for 45.25% of the total increase, and the carbon stock of forest land showed a fluctuating upward trend. 41.09% of the growth in the carbon stock of the entire ecosystem between 1990 and 2020 was accounted for by the carbon stock of construction land. As the area of construction land continues to rise, it is explained that the increase of carbon stock in construction land is due to the expansion of the construction land area, which is not the main factor of the increase of the total carbon stock in the Yunnan ecosystem. Water area has little to no effect on carbon stock and can be disregarded. As can be seen, the main driver of an increase in the ecological carbon stock in Yunnan province during the process of land use type shift is the expansion of forested land, whereas the main driver of a decline is the shrinkage of cropland. Although the land use pattern has changed, cropland, forest land, and grassland are still the main contributors to the ecosystem carbon stock, and the contribution of forest land and grassland to the ecosystem carbon stock has been increasing, while the contribution of cropland has decreased. The carbon stock of construction land has increased the most, but it has little effect on the enhancement of the total carbon stock of the province’s ecosystem.

### 3.5. Impact of Land Use Type Shift on Carbon Stock Change

The ArcGIS 10.2 overlay function was used to create the land use transfer layer, and the changes to the Yunnan ecosystem’s carbon stock were also documented in terms of land categorization. As shown in Figure 4, the carbon stocks of each land category are ranked as follows. forest land > grassland > cropland > others, and the carbon stocks of forest land account for 68.2% and 68.61% of the carbon stocks of the whole ecosystem from 1990 to 2020, respectively. Due to the shift of land use types, the proportion of carbon stock in forest land increases slightly. Furthermore, cropland carbon stocks account for 11.59% and 11.23% of total ecosystem carbon stocks in 1990 and 2020, respectively. In addition, 19.62% and 19.27% of the carbon stock of grassland account for the carbon stock of the whole ecosystem in 1990 and 2020, respectively. The shift of land use types did not play a significant role in the improvement of ecosystem carbon stock in Yunnan Province. From 1990–2020, the ecosystem carbon stock in Yunnan Province fluctuated and reached a peak around 2015, when the ecosystem carbon stock reached 8288.16 × 10^6^ t.

After 2015, it started to decline slowly. This has to do with how different land uses are distributed in Yunnan Province, one of the few regions in China with significant forested areas that preserves a significant amount of primary forests and woodlands. As a result, forest land has a substantially higher carbon density than other types of land use. Cropland, forests, and grasslands have exceptionally high carbon stocks, making up 11%, 68%, and 19% of the ecosystem’s total carbon stock, respectively. Nearly 70% of this contribution comes from diverse types of forest land use, demonstrating how effective forest vegetation is at sequestering carbon. As can be seen, the main factor contributing to the decrease in Yunnan Province’s ecosystem carbon stock during the process of land use type conversion was the shrinkage of cropland and grassland use areas, while the main factor contributing to the increase in ecosystem carbon stock was the expansion of forest land area. Following the change in land use, the main sources of ecosystem carbon stocks are still cropland, forests, and grasslands; however, the contributions of forests and grasslands to ecosystem carbon stocks are rising while those of cropland are declining. The province’s ecosystem’s overall carbon stock has improved, but it has minimal impact on the carbon stock of construction land, which has increased the most.

### 3.6. Land Use Type Area Projections for 2030

The accuracy test for the 2030 LUCC prediction for Yunnan Province was initially carried out by putting the 2000–2010 LUCC map data with a 10-year cycle into IDRISI. When the resulting simulated map of Yunnan Province in 2020 was compared to the real map, the kappa coefficient resulted in a value of 0.85 > 0.75, indicating high accuracy. The 2030 Yunnan LUCC simulation map was then predicted using the 2010 and 2020 Yunnan maps (Figure 2), and the 2020–2030 LUCC transfer matrix is displayed in Table 5.

The forecast of LUCC in 2020–2030 shows that the area of cropland, grassland, and unused land increases by 1206 km^2^, 18,294 km^2^, and 265 km^2^, respectively, while the area of forest land and water area decreases by 19,245 km^2^ and 821 km^2^, respectively, and the area of construction land remains basically unchanged. The land transfer matrix shows that the increase in grassland and cropland area mainly comes from the degradation of forest land, and it is predicted that 35.39% of the total area of grassland in 2030 will be transferred from forest land to grassland between 2020 and 2030. The net transfer of forest land accounts for 80.38% of the increase in grassland area in 2030, and the increase in cropland area comes from the transfer of forest land and grassland. The net transfer of forest land to cropland accounts for 18.33% of the cropland area in 2030. The increase in unused land area comes from grassland and forest land, and the total area transferred to unused land accounts for 55.33% of the unused land area in 2030.

## 4. Discussion

It is more efficient to examine regional carbon stocks and forecast future changes in carbon stocks using past ecosystem growth and trajectory changes [34,53]. The process of ecosystem change will result in equivalent changes in carbon stocks, and one of the most effective ways to monitor ecosystem changes is to analyze carbon stock fluctuations. In this study, the ecosystem carbon stocks were evaluated over five time periods between 1990 and 2020 using the InVEST model. The influence mechanisms of changing land use types on ecosystem carbon stocks were also analyzed. Finally, the CA-Markov model was combined to forecast the state of the land use pattern in 2030 and predict the future trend of ecosystem carbon stocks. This is crucial for the province of Yunnan’s scientific planning of land use patterns.

The carbon intensity of the six land types in Yunnan Province was assessed using the carbon stock module of the InVEST model in the following order. Forest land > grassland > cropland > water area > unused land > construction land. This is consistent with the study of Shi et al. [54] scholars. The majority of the aboveground biomass is released to the atmosphere as carbon when forest land is converted to grassland or farmland, while tree roots also decompose and release significant amounts of carbon [55]. According to earlier research [56,57,58,59], the primary cause of the drop in the ecosystem carbon stock in Yunnan Province has been the sharp reduction in forest land area. It emphasizes that implementing policies such as converting cropland to forests protects woodland areas, which has a big impact on improving ecosystems’ capacity to store carbon.

The carbon stock module of the InVEST model was used in this study to evaluate the Yunnan ecosystem’s carbon stock. The results of estimating carbon stocks in Yunnan ecosystems may not be accurate, despite the accuracy and science of the estimation process, because of the occurrence of ecosystem succession in Yunnan Province during the study period, the incorrect classification of land use types, and other uncertainties. Specifically, there are forest types in different successional stages in Yunnan Province, such as primary forest to early successional forest, which are not measured differently. The inaccuracy of land use type classification also leads to errors in the measurement results. Additionally, because the model relies on data on the carbon densities of various land types in the process of valuing carbon stocks, and because the model assumes that the densities of various land types are fixed over time, the values of carbon densities are primarily derived by consulting earlier studies. Despite the fact that the relationship between the precipitation factor, mean annual temperature, and ecosystem carbon stock has been corrected, there are still errors in the values of carbon densities. Future research should take into account factors such as primary forest masking, increase the resolution of forest types during the mapping process, and conduct long-term dynamic monitoring of the effects of temperature, photosynthetic rate, soil microbial changes, and human activities on soil carbon density. Finally, in terms of improving the estimation and prediction of carbon stocks, future studies should take the aforementioned concerns into account. The CA-Markov model’s prediction of land use types in Yunnan Province in 2030 does not accurately account for variables such as urban development scenarios and ecological protection scenarios.

## 5. Conclusions

With the use of the carbon stock module of the InVEST model, this study examined the spatial and temporal changes in land use patterns in Yunnan Province from 1990 to 2020 as well as the spatial and chronological changes in ecosystem carbon stocks in Yunnan Province from 1990 to 2020. The trends of LUCC and ecosystem carbon stocks in Yunnan Province in 2030 were predicted by integrating the CA-Markov model with analysis of the changes in ecosystem carbon stocks brought on by variations in land use types. The following conclusions were reached:

(1) In Yunnan Province, between 1990 and 2020, 97% of the province’s land area was used for these three land use types: forest, grassland, and arable land. Over 30 years, grassland, arable land, and unoccupied land saw net outflows among the six land types, but building land, forest land and water areas experienced net inflows;

(2) Although the overall ecosystem carbon stock in Yunnan Province essentially remained constant throughout the period of five time points from 1990 to 2020, the change and contribution of the carbon stock varied among diverse land types. Since forests make up the majority of these land uses, it is clear that they have a greater capacity to store carbon than other types of plants. While the carbon stock of forest land increased the most, making up 45.25% of the overall increase, the carbon stock of grassland decreased the most and kept declining;

(3) The impact of different land use types on ecosystem carbon stocks revealed that Yunnan Province’s ecosystem carbon stocks did not significantly improve when land use types changed. This has to do with how different land uses are distributed in Yunnan Province, where a lot of primary forests and woodlands have been conserved, giving woodlands a significantly higher carbon density than other land uses. In Yunnan Province, the loss in ecological carbon stock is primarily attributable to the shrinkage of arable land and grassland area, while a gain in ecological carbon stock is primarily attributable to the expansion of forest land area;

(4) The area of arable land, grassland, and unused land will increase, the area of forest land and water area will show a reduction, and the area of construction land will essentially stays the same, according to the LUCC’s estimate for Yunnan Province from 2020 to 2030 using the CA-Markov model. According to the impact of different land use types on ecosystem carbon stocks between 1990 and 2020, Yunnan Province’s total ecosystem carbon stock will gradually decline over the next ten years, despite an increase in the amount of cultivated land and grassland. The main factor contributing to this decline is the shrinking amount of forest land, which has a negative impact on the carbon stock of that land.

## Figures and Tables

**Figure 1 ijerph-19-16059-f001:**
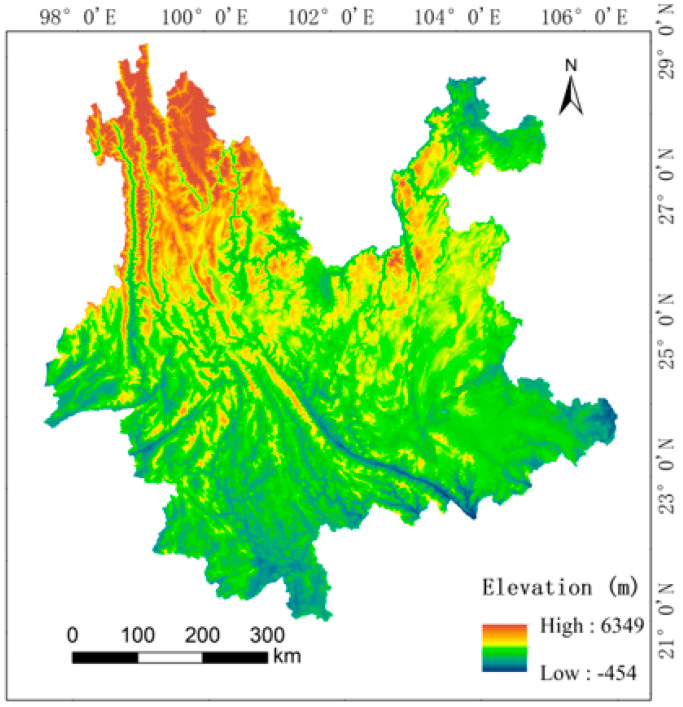
Overview of the study area.

**Figure 2 ijerph-19-16059-f002:**
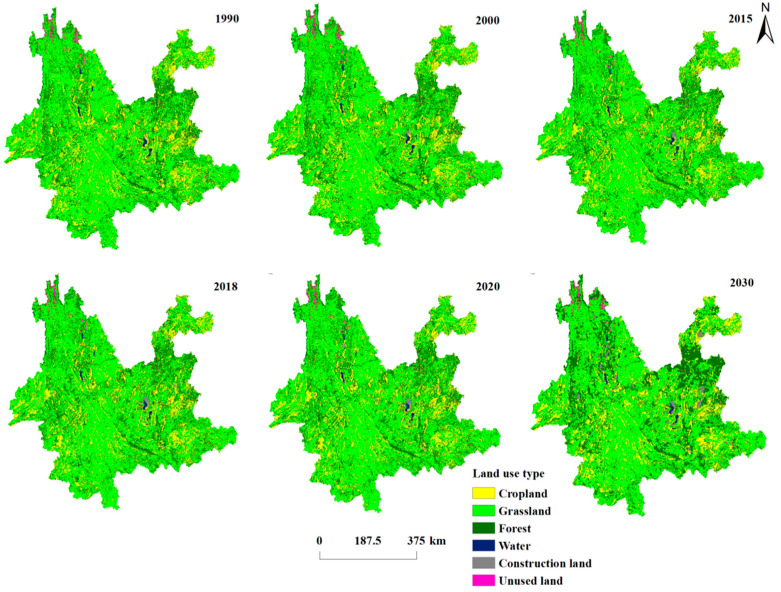
Spatial distribution of land use types in Yunnan Province, 1990–2030.

**Figure 3 ijerph-19-16059-f003:**
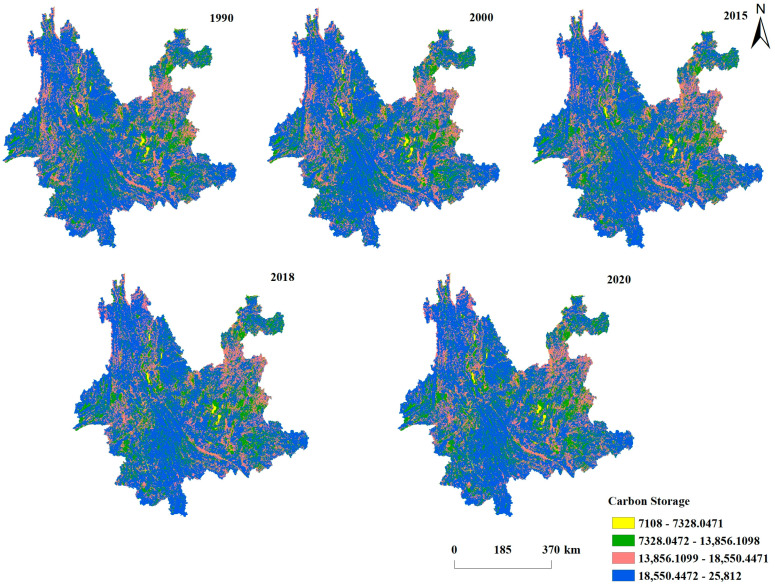
Spatial and temporal distribution of ecosystem carbon stocks in Yunnan Province from 1990 to 2020.

**Figure 4 ijerph-19-16059-f004:**
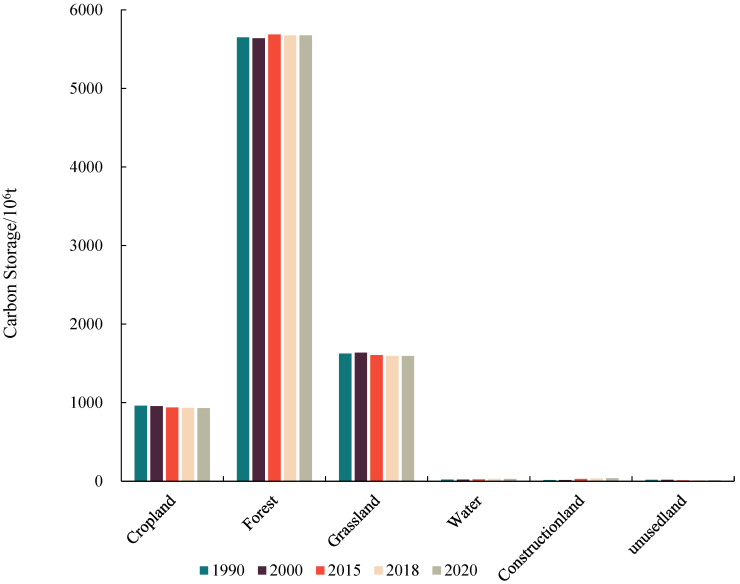
Carbon stock share of ecosystems in Yunnan Province from 1990 to 2020.

**Table 1 ijerph-19-16059-t001:** Carbon density of different land use types in Yunnan Province (t/hm^2^).

Land Use Type	Above-Ground Carbon Density	Subsurface Carbon Density	Soil Carbon Density	Carbon Density of Dead Organic Matter
Cropland	27.89	62	47.52	1
Forest land	48.18	131.7	49.24	29
Grassland	38.24	95.6	50.45	1
Water area	27.55	0	43.53	0
Construction Land	26.43	0	47.48	0
Unused land	30.23	0	40.75	3

**Table 2 ijerph-19-16059-t002:** Land use area and proportion in Yunnan Province from 1990 to 2020 (km^2^).

Land Use Type	1990		2000		2015		2018		2020	
	Area	%	Area	%	Area	%	Area	%	Area	%
Cropland	69,371	18.13	69,001	18.03	67,688	17.67	67,247	17.55	67,105	17.52
Forest land	218,881	57.2	218,421	57.08	220,225	57.49	219,869	57.4	219,823	57.38
Grassland	87,732	22.9	88,292	23.07	86,586	22.6	86,051	22.46	85,995	22.45
Water area	2805	0.73	2831	0.74	3239	0.85	3843	1	3838	1
Construction Land	1788	0.47	2031	0.53	3777	0.99	4513	1.18	4775	1.25
Unused land	2105	0.55	2105	0.55	1561	0.41	1553	0.41	1540	0.4

**Table 3 ijerph-19-16059-t003:** Land use transfer matrix of Yunnan Province from 1990 to 2020 (km^2^).

1990	2020						
	Grassland	Cropland	Construction Land	Forest Land	Water Area	Unused Land	Total
Grassland	44,340.97	12,648.76	801.74	28,672.68	782.59	307.26	87,554.00
Cropland	12,076.72	31,707.80	2551.25	22,223.81	690.02	49.10	69,298.69
Construction land	153.05	839.59	560.42	187.75	40.58	5.02	1786.40
Forest land	27,970.77	21,188.40	764.68	167,415.10	937.41	296.13	218,572.51
Water area	393.49	556.60	88.91	427.43	1315.71	13.22	2795.35
Unused land	806.10	63.81	2.89	335.04	24.85	853.64	2086.32
Total	85,741.10	67,004.96	4769.88	219,261.81	3791.16	1524.36	382,093.26

**Table 4 ijerph-19-16059-t004:** Carbon stocks of different land use types in Yunnan Province, 1990–2020 (10^6^ t).

Land Use Type	1990	2000	2015	2018	2020
Cropland	960.16	955.04	936.87	930.77	928.80
Forest land	5649.76	5637.88	5684.45	5675.26	5674.07
Grassland	1625.59	1635.96	1604.35	1594.44	1593.40
Water area	19.94	20.12	23.02	27.32	27.28
Construction Land	13.22	15.01	27.92	33.36	35.29
Unused land	15.57	15.57	11.55	11.49	11.39

**Table 5 ijerph-19-16059-t005:** Land use transfer matrix of Yunnan Province from 2020 to 2030 (km^2^).

2030	2020						
	Grassland	Cropland	Construction Land	Forest Land	Water Area	Unused Land	Total
Grass land	50,912.78	14,217.83	871.75	36,907.57	850.93	345.90	104,106.76
Crop land	11,122.83	31,070.15	1695.01	23,641.68	638.83	68.80	68,237.30
Construction land	332.14	2436.57	1444.72	342.58	96.29	38.44	4690.74
Forest land	22,203.35	18,651.90	633.32	157,653.31	864.14	268.41	200,274.43
Water area	438.06	594.51	124.14	449.05	1339.49	50.11	2995.35
Unused land	731.90	33.99	0.95	266.83	1.48	752.71	1787.86
Total	85,741.05	67,004.96	4769.88	219,261.03	3791.16	1524.36	382,092.44

## Data Availability

The datasets provided in the study are included in the article materials. For further enquiries, please contact the corresponding author.

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
