# Peer review of "Spatial and Temporal Variation and Prediction of Ecosystem Carbon Stocks in Yunnan Province Based on Land Use Change"

_ijerph, 2022, doi:10.3390/ijerph192316059_

Round 1

Reviewer 1 Report

Review for Manuscript:

Spatial and temporal variation and prediction of ecosystem carbon stocks in Yunnan Province based on land use change

The manuscript submitted provides a robust and well-organized review of its topics of interest. The introduction introduces the topics of interest well, including making clear and informed arguments about the benefits of natural ecosystems as carbon sinks as well as recent research surrounding carbon assessment research. The study area for the research questions is also solid with Yunnan having the second-highest forest cover in China and complex land-use and land-cover change dynamics of local-to-global importance. Overall, the study provides a useful and detailed case study of carbon and LUCC dynamics in Yunnan during a crucial period of change (1990-2020). Some specific suggestions and questions are detailed below:

You say in lines 94-95:  “For instance, Zhao et al. [36] examined the geographical and temporal variations of the carbon stock in the study and utilized the InVEST model to account for the carbon stock in the Heihe River Basin.” What did they find? Also, you mention next “there are still several shortcomings in the previous studies.” While you cite three case studies, you only mention one, “Zhao et al. [36].” I suggest you mention another study and perhaps an outcome or two of these studies to further support your argument about the value and rigor of the InVEST approach.

You say in line 110 that “The analysis of carbon stock changes is one of the effective techniques to regulate ecosystem changes.” Is it “regulate” or “monitor?” On lines 347-348 you also say, “The process of ecosystem change will result in equivalent changes in carbon stocks, and one of the most effective ways to control ecosystem changes is to analyze carbon stock fluctuations.” Again, I don’t think you mean to say “control”? “Monitor” seems a more accurate word to use here. I would see enforcement tools, like fines, as “control.”

Line 114 – You say, “Geographically speaking, Yunnan Province is in southwest China. 39.41 million hectares are included in the overall area.” Make one sentence here to improve sentence flow. I suggest, “Geographically speaking, Yunnan Province is in southwest China and covers 39.41 million hectares.”

Line 117 - “With 25,994,400 hm2 of woodland,” Suggest you use hectares (ha). You also use km2 in some other places in the manuscript. I suggest you make all area reporting uniform in the manuscript in ha or km2

Line 118-128. You say basically the same thing for 9 lines. I suggest cutting down the details supporting the point that Yunnan is forest rich compared to other parts of China.

Line 145-146 – You say, “stocks in Yunnan Province from 1990 to 2020 using the carbon stock module in the InVEST 3.12.0 model. must take into account every land use-related carbon sink, such as the soil 1carbon sink, the above-ground biomass carbon sink, and the below-ground biomass carbon sink.” The second sentence starts poorly worded. Should it start with “InVEST”?

Lines 147-156 - It is not as clear as it could be where you obtained your carbon storage estimates. You discuss the formulas used to correct your carbon density data, but it’s not clear what the “pertinent literature” was that you obtained the density estimates. You need more detail here, otherwise, we cannot judge the quality of the dataset(s) used.

It would be worthwhile to note the software you used to run the CA-Markov model. Was it also IDRSI?

Line 372-375 – I suggest you make the caveat somewhere in this paper that your carbon storage estimates are likely to be imprecise due to classification errors and ecosystem successional stages. Specifically, you report a generic forest cover classification carbon storage value and land-change cover, but you also note in the study that in Yunnan there are forest types at different successional stages, such as primary forests to early successional forests. You could mention this caveat at the end of your discussion where you mention other uncertainties that require future study to improve carbon storage estimates. Relatedly, future research could include an approach to increase the resolution of forest type at the mapping stage, such as primary forest masking.

Line 327 – You say, “…indicating accurate accuracy.” Suggest “high accuracy”

Line 344 – You start off the section with the sentence, “More efficient methods include regional carbon stock analysis and future carbon stock change prediction using historical ecosystem growth and trajectory changes [46, 47]. 346” Be more precise. What kind of methods are you referring to?

Line 361 – You say, “According to earlier researchers' results [50–53], the main cause of the fast decline in forest land area is the loss in carbon stock in the Yunnan ecosystem.” I don’t think this sentence is written in English correctly. What is the main driver of forest change? Loss of carbon stocks would be an outcome of the drivers, not the driver.

Line 386 - “The next conclusions were reached:” Should be: “The following conclusions were reached:”

381- 443 - The conclusions could be made more concise. You could remove half the content and be okay in my view. Everything written is largely a restatement of what was written above. Focus here less on the data/results and instead summarize the data and remind the reader of the policy/science implications. As written, it is too dense to extract much value from a policy perspective.

Reviewer 2 Report

Overall, the manuscript is well prepared. The introduction part provided sufficient background information. The methodology section needs to be improved. The result section is also well written and the conclusions are supported by the results.

Please revise the manuscript according to the following comments-

1. The objective of this research is not clearly mentioned anywhere. I suggest adding a paragraph to clarify what is your research question and hypothesis.

2. Figure 1: Add geographic information (latitude-longitude).  The map is showing elevation information. How is that related to the context of the research? Instead, you can use a LULC map or another appropriate one.

3. Describe the models (i.e., InVEST, CA-Markov) in more detail (i.e., what parameters were used, have you modified the model, add any necessary figures) so that the results can be reproduced.

4. Figure 2: Increase all font sizes. Grasslands and Forests are difficult to differentiate because of their similar color. Use a contrasting color.

5. Figure 4: Increase all font sizes. Add a unit for carbon storage. Instead of similar colors, use contrasting colors (i.e., blue, red, green, yellow) to make the changes more visible.

Reviewer 3 Report

My understanding of this article is that you used InVEST model to study carbon stocks in Yunnan Province at different periods and analyzed the relationship between land use type shift and carbon stock. Through this relationship, the type of land use in 2030 was predicted with CA-Markov model. I have some questions, could you please state them?

1. What is the relationship between carbon stocks and carbon sinks? Are carbon stocks the result of carbon sinks?

2.Page 1, lines 40 to 42, you say that we must balance the two factors (carbon sources and carbon sinks) in the emission reduction process, right? But did you consider carbon sinks in your article? If you do, please state it?

3.Page 3, line 101, you say that the previous studies lack regional studies, but the second example you gave above is regional study. Hope you can narrate more accurate.

4.Page 4, line 140, what does 1:1000000 refer to, a map scale or something else?

5.Page 7, lines 223 to 227, I do not understand this two sentences, what are the transferred out area and transferred area?

6.Page 10, lines 277 to 279, why the extension of construction area is not the cause of the growth in the carbon stock of construction land?

7. Page 12, line 380, the conclusion is too wordy, and some of the data is repeated above.
